# A Low-Complexity Accurate Ranging Algorithm for a Switch Machine Working Component Based on the Mask RCNN

**Lili Wei, Lingkai Kong, Zhigang Liu, Zhenglong Yang \* and Hua Zhang**

School of Urban Railway Transportation, Shanghai University of Engineering Science, Shanghai 201620, China; weillwzm@126.com (L.W.); klk6ge8@163.com (L.K.); liuzhigang@sues.edu.cn (Z.L.); zhxueju@163.com (H.Z.)
\* Correspondence: 10190004@sues.edu.cn

**Abstract:** According to the intelligent development needs of railway operation and maintenance, turnout maintenance also needs an efficient and intelligent means of detection. It is the main method used to measure the access depth of static contact manually. In order to change the disadvantages of the low efficiency and strong subjectivity of traditional schemes, a low-complexity accurate ranging algorithm of the Mask RCNN is proposed to measure the on–off working parts. Firstly, the Mask RCNN and an interactive iterative method are used to segment the region of interest accurately twice. Secondly, the graph distortion is corrected according to the vertex mapping principle. Finally, the accurate actual distance is calculated through fitting the linear distance transformation equation. Through the secondary segmentation and correction algorithm, the accurate calculation of a small target is completed. The experimental results show that the algorithm can accurately measure the distance of different working parts; the average processing time is 0.8 s/amplitude and the measurement error is ±1 mm.

**Keywords:** Mask RCNN; image segmentation; 3D correction; equal scaling

## 1. Introduction

Switch machines play an important role in the safe operation of rail transit. Once the switch machine stops working correctly, it will affect the operating efficiency and in the smallest cases can cause train derailment accidents. The switch machine will inevitably suffer from wear and aging problems in daily use, so regular maintenance is necessary. The access depth between the dynamic and static contact group of the automatic switch circuit controller determines the working situation of the switch machine.

A set of automatic opening and closing devices and shown in Figure 1, which are connected by moving the contact up and down to connect and energize the upper or lower points. The purple area at the top and bottom of the automatic switch is the static contact area, which mainly consists of three sets of elastic spring contacts and are responsible for connecting the circuit for work. The small solid green circle is the moving contact of the automatic switch. It swings up and down with the movement of the base and contacts the elastic spring in the static contact area, conducting current.

If the access depth does not meet the standard requirements, it will lead to circuit failure and cause serious accidents. At present, the inspection of the switch circuit controller still requires manual participation. Due to the complexity of the switch machine structure and the actual working environment, it is difficult to obtain an accurate measurement result. On the other hand, the subjective experience will also affect the final judgement.

Deep learning technology has made great improvements to the automatic measurement of the switch machine [1]. Normally, there are many typical automatic measurement methods, such as the ultrasonic detection methods [2,3] and the laser detection method [4]. However, the ultrasonic and laser detection methods require the surface of the measured object to have high reflectivity. If it does not meet this standard, the measurement will be

invalid. With the development of deep learning, the convolutional neural network can extract the characteristics of the object efficiently and give a good prediction [5].

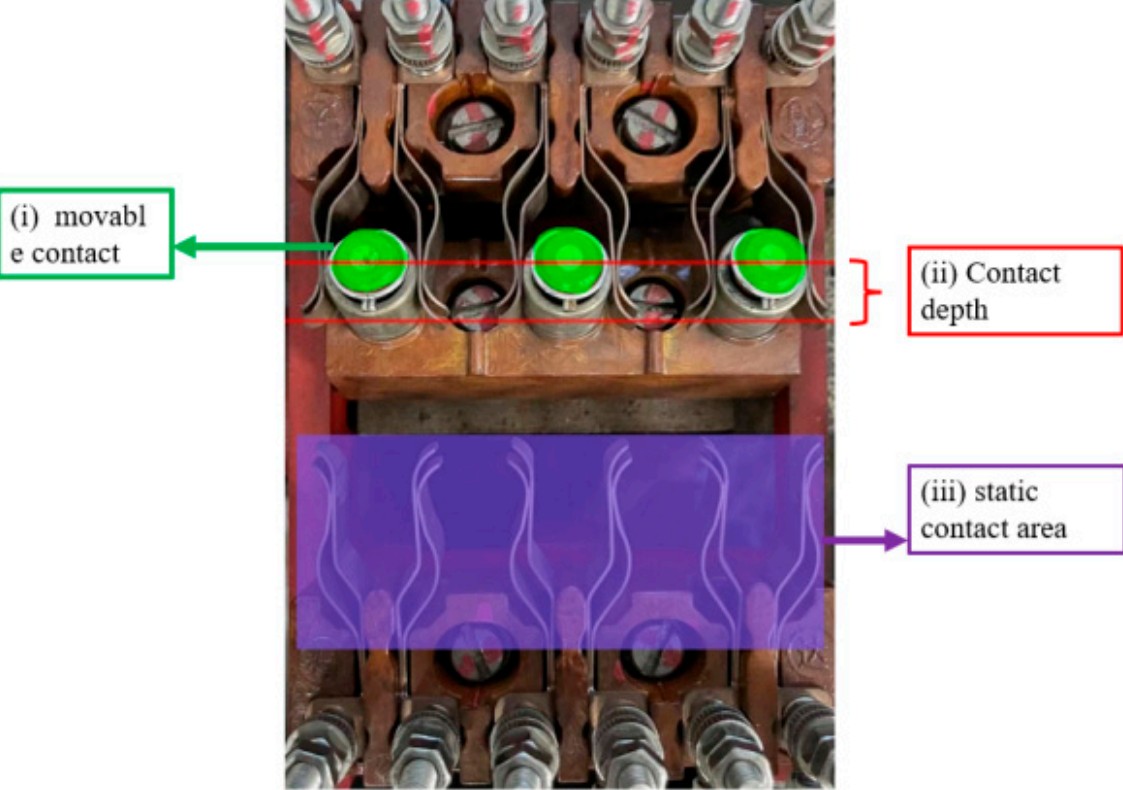

**Figure 1.** Automatic switch machine operating mechanism.

With the development of deep learning technology, deep-learning-based machine vision target positioning methods have been rapidly developed and applied in various fields. Both the Yolo (You Only Look Once) object segmentation algorithm proposed at the Machine Vision Conference for the first time in 2016 and the regional convolutional neural network (RCNN) based on the convolutional neural network (CNN) have led the rapid development of object detection. Then, the Fast RCNN, Faster RCNN, and Mask RCNN have been proposed one after another. The Mask RCNN reduces the error of image compression and improves the segmentation accuracy by adding RoI-Align instead of RoI-Pooling. On the other hand, target detection and target segmentation are performed synchronously by adding mask branches. It is currently the most advanced algorithm in the direction of the RCNN.

Zhang et al. [6] designed a BP neural network and PNN to diagnose switch faults. Jia et al. [7] used the deep neural networks for the deficiencies. Li Chao et al. [8] detected the gap of the switch machine based on the canny operator and obtained the position of the gap through the edge detection algorithm; however, it is only useful for the fixed type of switch machine, which will be hard to use in practice. Tao et al. [9] used the Yolo network to detect the gap of the switch machine. The detection method for the gap has made significant progress in terms of feasibility; however, this function is not scalable. ZHANG et al. [10] improved the machine vision method for object detection. However, the complex detection scene will affect the measurement results. The Mask RCNN [11] adds a new branch network for generating the segmentation mask to the region of interest (ROI). Yin [12] et al. improved the positioning accuracy through two Mask RCNN networks. However, the proposed algorithm worked well in the experimental environment. When the working condition changes in a complicated way, the predicted results are unsuitable for precise positioning.

Different from the previous works, the proposed algorithm is suitable for multiple types of switch machines and different working environments. On the other hand, the low complex and high precise ranging are the outstanding abilities of the proposed algorithm. To measure the access depth of the dynamic contact group, there are mainly three steps: 1. Target detection and fine segmentation. 2. Three-dimensional correction. 3. The linear fitting and transformation of the distance. The experimental results show that the proposed method can reduce or even eliminate the interference caused by complex environments to measure the access depth.

## 2. Proposed Algorithm

### 2.1. Mask RCNN

From the network structure, the Mask RCNN can be roughly divided into convolutional backbone, RPN, feature map, ROI-Align layer, fixed size feature map, mask branch, fully connected layers, box regression, and classification. Such as Figure 2 depicts the architecture diagram of the MASKRCCC network.

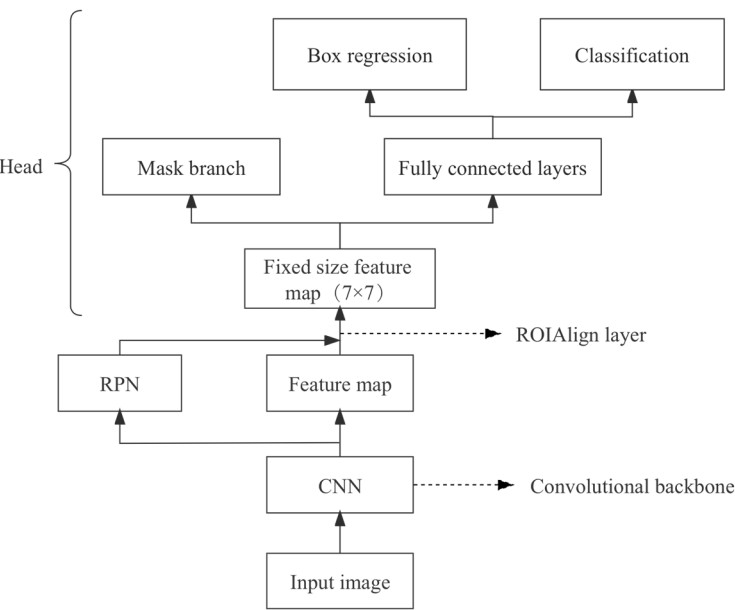

**Figure 2.** Mask RCNN network structure.

Compared with other RCNN segmentation networks, the RoI-Align layer is employed to accurately align RoI features, which effectively improves the accuracy of detection and segmentation. Moreover, the mask branch is introduced to perform binary classification on each pixel within RoIs, achieving the precise segmentation of object instances. The Mask RCNN has demonstrated good performance on various benchmark datasets, and has the advantages of high accuracy, efficiency, scalability, flexibility, and compatibility, making it one of the leading models in the field of object detection and instance segmentation.

### 2.2. Second Precise Segmentation Based on the Mask RCNN

We use the Mask RCNN model to predict the position of the screws on the dynamic and static contact groups of the switch machine. The results are shown in Figure 3. Figure 3a shows the predicted positions of the screws from a mobile phone image. In Figure 3b, the predicted screw positions are converted into a black-and-white image.

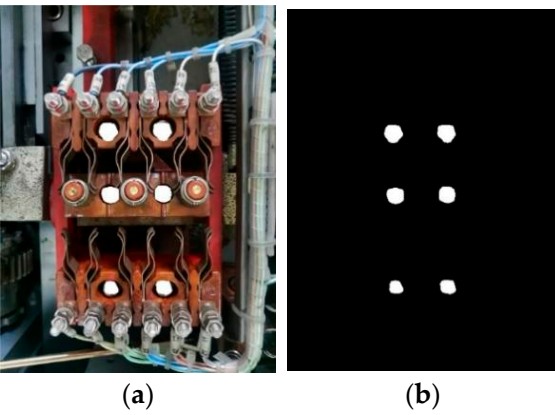

**Figure 3.** Prediction results of the Mask RCNN for the screws. (**a**) Prediction results in original image; (**b**) Separate prediction result.

From Figure 3, we can see that the predicted mask of the screws are rough, which will affect the precise segmentation of the screws. Therefore, the predicted mask of the screws needs a second segmentation for obtaining the complete screw regions. For the second segmentation, we select the predicted mask from the Mask RCNN as the reference. Then, the near regions of the predicted mask will be explored through a Gaussian mixture model (GMM) [13] to determine the final screw regions. The second segmentation results are shown in Figure 4.

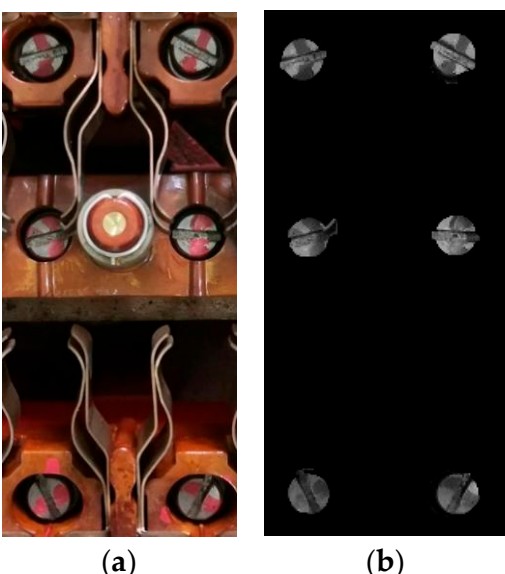

**Figure 4.** Second segmentation. (**a**) Second seg-mentation in original image; (**b**) Separate second segmentation.

Figure 4a shows the actual screw images, and Figure 4b shows the second segmentation results from Figure 4a. Compared with Figure 3, we can see that the screws can be segmented precisely.

According to the second segmentation, by performing the minimum complement circle of the screws, the minimum-area enclosing circle can be modeled, which is shown in Figure 5.

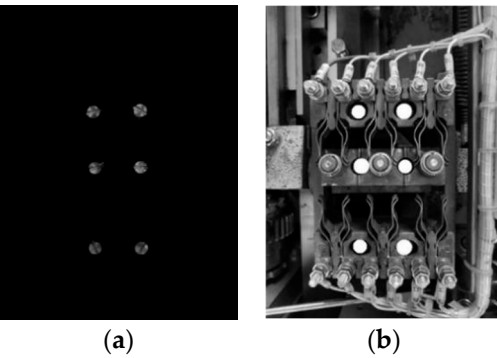

(**a**)　　　　　　(**b**)

**Figure 5.** Minimum-area enclosing circle of the screws. (**a**) Separate second segmentation; (**b**) Minimum-area enclosing circle of the screws in original image.

Figure 5a shows the second segmentation of the screws and Figure 5b shows the minimum-area enclosing circle of Figure 5a. From Figure 5, we can see that the minimum-area enclosing circle can cover the screws very well. Then, we can use the rectangular region composed by the minimum-area enclosing circles for the distortion correction.

*2.3. Distortion Correction Based on Affine Mapping*

Due to the image being captured with a camera or mobile phone, the deformation of the switch machine may occur. Therefore, the position of the minimum-area enclosing circle of the screws in the image may deviate from the actual position. We use affine mapping to reduce the distortion and correct the position of the minimum-area enclosing circle.

For the distortion correction, six key points, which are shown in Figure 6 (A, B, C, D, E and F), are used to map the distorted image to the corrected image.

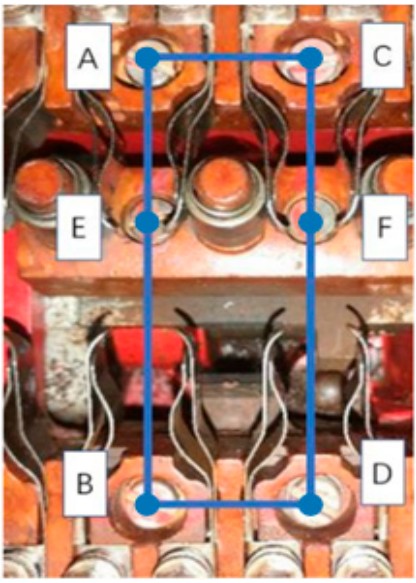

**Figure 6.** Key points for affine mapping.

In Figure 6, the coordinates of A, B, C, D, E, and F are (a1,a2), (b1,b2), (c1,c2), (d1,d2), (e1,e2), and (f1,f2), respectively. The distance of AC is:

$$AC = L = \sqrt{(a1 - c1)^2 + (a2 - c2)^2}$$

As the actual proportion of AC and CD is 3:4, the calculated distance of CD is:

$$CD = H = L \cdot \frac{4}{3}$$

The calculated coordinates of B, C, and D are $B^*$, $C^*$, and $D^*$, which are (a1, a2 + H), (a1 + L, a2), and (a1 + L, a2 + H), respectively. Then, the affine mapping function is modeled using:

$$\begin{bmatrix} x \\ y \\ z \end{bmatrix} = \begin{bmatrix} a_{11} & a_{12} & a_{13} \\ a_{21} & a_{22} & a_{23} \\ a_{31} & a_{32} & a_{33} \end{bmatrix} \begin{bmatrix} u \\ v \\ 1 \end{bmatrix} \tag{1}$$

Therefore, Equation (1) can be converted to:

$$x' = \frac{x}{z} = \frac{a_{11}u + a_{12}v + a_{13}}{a_{31}u + a_{32}v + a_{33}} \tag{2}$$

$$y' = \frac{y}{z} = \frac{a_{21}u + a_{22}v + a_{23}}{a_{31}u + a_{32}v + a_{33}} \tag{3}$$

where $(x', y')$ are the corrected pixels and $(u, v)$ are the original pixels. $\begin{bmatrix} a_{11} & a_{12} & a_{13} \\ a_{21} & a_{22} & a_{23} \\ a_{31} & a_{32} & a_{33} \end{bmatrix}$ is

the perspective transformation affine matrix. Taking A, $B^*$, $C^*$, and $D^*$ and A, B, C, and D into $(x', y')$ and $(u, v)$, respectively, the solution of the perspective transformation affine matrix can be obtained. The distortion correction results are shown in Figure 7.

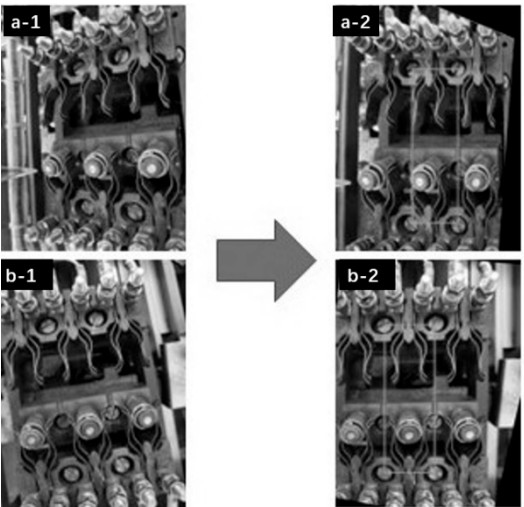

**Figure 7.** Distortion correction of the switch machine. (**a-1**,**b-1**) are original drawing; (**a-2**,**b-2**) are corrected images

From Figure 7(a-1,a-2), it can be seen that the switch machine in Figure 7a shows deformation in a three-dimensional direction. However, in Figure 7b, the deformation is corrected extremely. Similarly, in Figure 7(b-1), the deformation of the switch machine is mainly in the horizontal and vertical direction, and the corrected result in Figure 7(b-2) is almost same as the actual one.

### 2.4. Distance Linear Fitting and Transformation

We measure the six distance points of the screws and the corresponding dynamic contact group, which are *b*, *c*, *d*, *e*, *f*, and *g* and *h*, *i*, *j*, *k*, *l*, and *m*, respectively in Figure 8. The moving distances of the screws, which are *bc*, *bd*, *be*, *bf*, and *bg*, and the corresponding moving distances of the dynamic contact group, which are *hi*, *hj*, *hk*, *hl*, and *hm*, are used

to fit the moving function of the screws in the static contact group and the dynamic contact group.

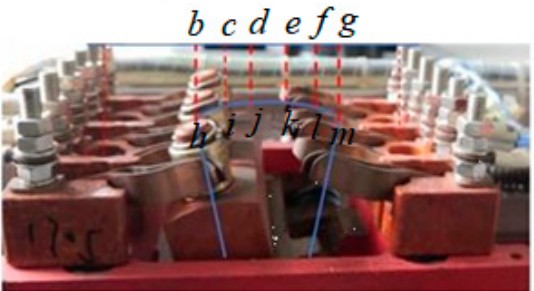

**Figure 8.** Distance transformation of the dynamic and static contact group.

From Figure 8, the access depth between the static and the dynamic contact group can be transformed with a third-order function, which is:

$$y = \alpha x^3 + \beta x^2 + \gamma x^1 + \delta \tag{4}$$

where $y$ is the access depth of the dynamic contact group, $x$ is the moving distance of the screws in the dynamic contact group, and $\alpha$, $\beta$, $\gamma$, and $\delta$ are the coefficients obtained via the fitting progress. Then, the moving distance of the screws in the dynamic contact group can be obtained with:

$$\frac{op}{qr} = \frac{op_{actual}}{qr_{actual}} \tag{5}$$

where $op$ is the distance of the two static contact groups in Figure 9, $op_{actual}$ is the actual distance in the real world, $qr$ is the moving distance of the screws in the dynamic contact group, and $qr_{actual}$ is the actual distance in the real world. In Equation (5), $op$ and $qr$ can be obtained in Section 2.2, and $op_{actual}$ can be measured by hand. Therefore, $qr_{actual}$ can be calculated with Equation (5), and then, by taking $qr_{actual}$ into Equation (4), the actual access depth can be obtained.

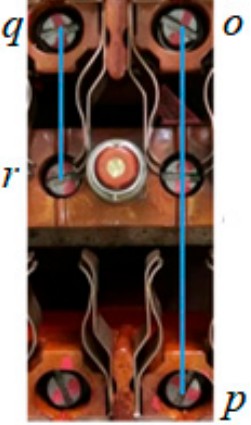

**Figure 9.** Moving distance of the screws in the dynamic contact group.

### 2.5. Overall Process of the Proposed Algorithm

The overall process of the proposed algorithm is introduced in this section, which is shown in Algorithm 1.

---

**Algorithm 1.** Overall process of the pseudo code algorithm.

---

1. Begin
2. Input image
3. If image_width>image_height
　　Rotate image to 90°
　　Crop image to 1280 × 960
　Else
　　Crop image to 1280 × 960
4. Input image into MaskRCNN
5. Output Mask
6. Take Mask into second segmentation
7. Get new Mask
8. Correct distortion based on affine mapping with new Mask
9. Distance transformation with linear fitting function
10. Output access depth
11. End

---

For the process in Algorithm 1, the image rotating operation will adjust the direction of the image to input into the Mask RCNN network. For the network output, the mask will be generated, and then the second segmentation is performed. According to the segmentation, the screw regions have been obtained with high confidence, and the center position of the screws can be confirmed. Since the input image may be deformed, the six center positions of the screws will be used for the affine mapping operation to reduce the distortion of the image. Finally, the access depth of the dynamic and static contact group in the image will be calculated, and the actual access depth will be transformed using the proportional scale method.

## 3. Experimental Results

### 3.1. Introduction of the Evaluation Method and the Experiments

For the experiments, the actual access depth of the dynamic and static contact group is measured by hand using the vernier scale. The access depth accuracy of the dynamic and static contact group is defined as:

$$T = \left| T_{algorithm} - T_{actual} \right| \tag{6}$$

where $T_{algorithm}$ is the distance calculated from the proposed algorithm and $T_{actual}$ is the actual access depth.

For the evaluation of the segmentation accuracy, *IoU* and *Dice*, shown in Figure 10, are used. The calculation formulas are:

$$IoU = \frac{T \cap P}{T \cup P} = \frac{Tp}{FP + TP + FN} \tag{7}$$

$$Dice = \frac{2|T \cap P|}{|T| + |P|} = \frac{2TP}{FP + 2 * TP + FN} \tag{8}$$

where *TP* is True Positive, *TN* is True Negative, *FP* is False Positive, and *FN* is False Negative.

The platform of the experiment is: Windows 10 Professional 64-bit, Intel i7-8700 @ 3.2 GHz, NVIDIA RTX 3080, Tensorflow-gpu 2.5.0, 32 G RAM, Python 3.6. All the images of the switch machines are obtained from China Shanghai Shentong Metro Co., Ltd., and then the images are rotated and cropped to the fixed size of 1280 × 960. There are six types of switch machines which are shown in Figure 11 in the dataset. The number of images in the dataset for training is 2400, in which the training dataset contains 400 images of each kind of switch machine. There are 600 images for testing, in which each kind of switch machine has 100 images. The coco [14] model is used to initialize the network.

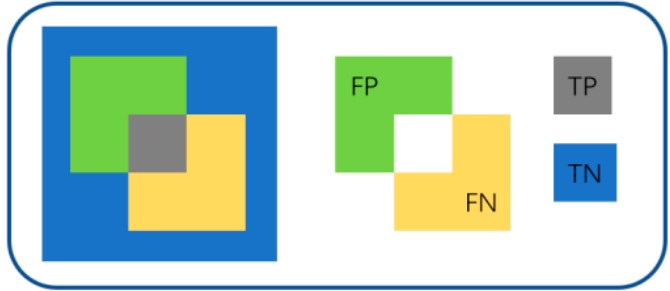

**Figure 10.** Segmentation description.

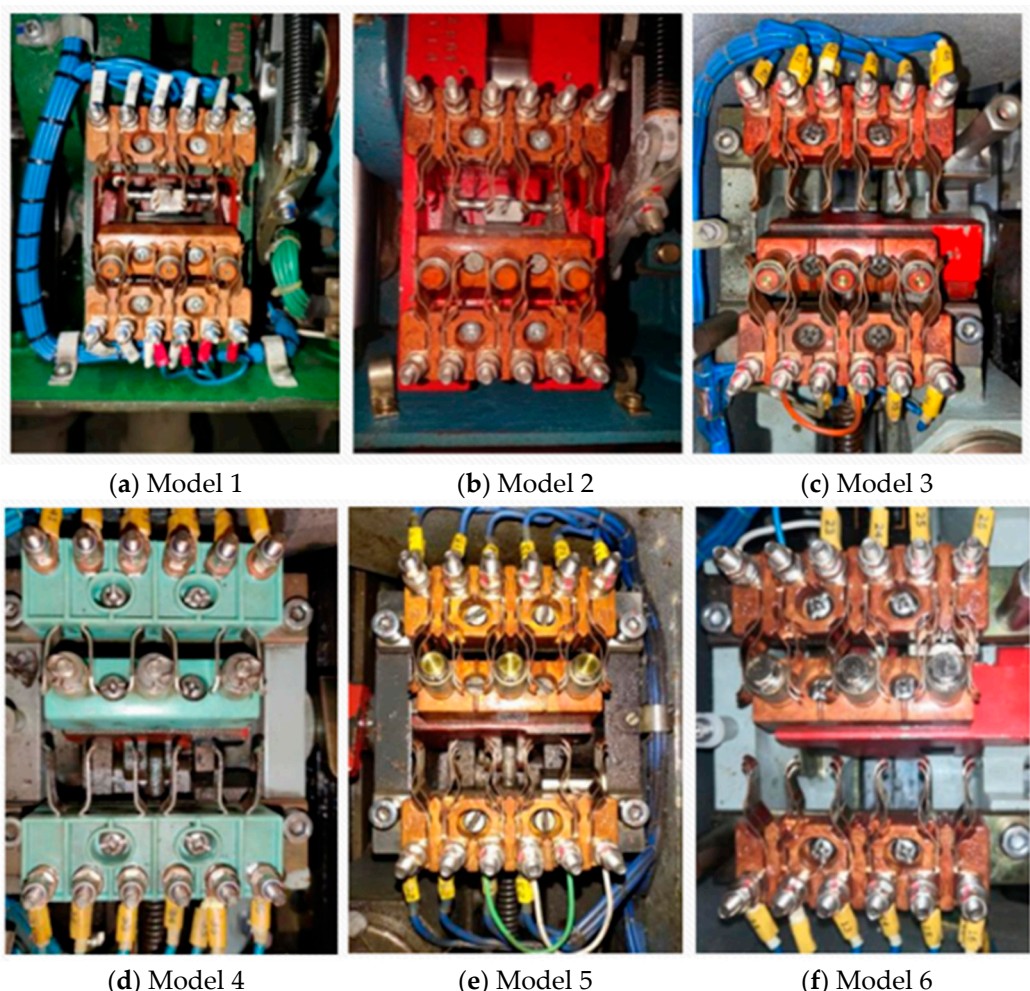

| (**a**) Model 1 | (**b**) Model 2 | (**c**) Model 3 |
|---|---|---|
| (**d**) Model 4 | (**e**) Model 5 | (**f**) Model 6 |

**Figure 11.** Six types of switch machines.

### *3.2. Training Results of the Mask RCNN Network*

The multitask loss curves of the Mask RCNN are shown in Figure 12. In Figure 12, the X-axis represents the number of training iterations, and the Y-axis represents the value of loss results. Four kinds of loss results are used to evaluate the performance of the training, which are total loss, class loss, mask loss, and bounding box regression loss, respectively. Total loss is the main loss function of the Mask RCNN, which is used to measure the overall performance of the network. Class loss is used to measure the prediction accuracy of the network to the target object category. It calculates the loss by comparing the difference between the prediction result and the label. Mask loss is used to measure the accuracy of the pixel-level segmentation of the target object. It calculates the loss by comparing the difference between the mask prediction results of the network and the real mask. Bounding

box regression (bbox) loss is used to measure the prediction accuracy of the bounding box of the target. Based on the smooth L1 function, it calculates the loss by comparing the predicted results of bounding box with the real bounding box. The bbox loss is to minimize the difference between the predicted bounding box and the real bounding box. These kinds of the loss results are shown in Figure 12a–d, respectively. From Figure 12, it can be seen that a sudden decreasing occurs during the training process; however, the overall trend is downward. This is usually caused by the transition from the training head to the training full data or the high learning rate of the model. Although there are fluctuations in the four losses, the training losses gradually reduce. In other words, the training performance of the network gradually improves. The recognition rate of the training results are shown in Table 1.

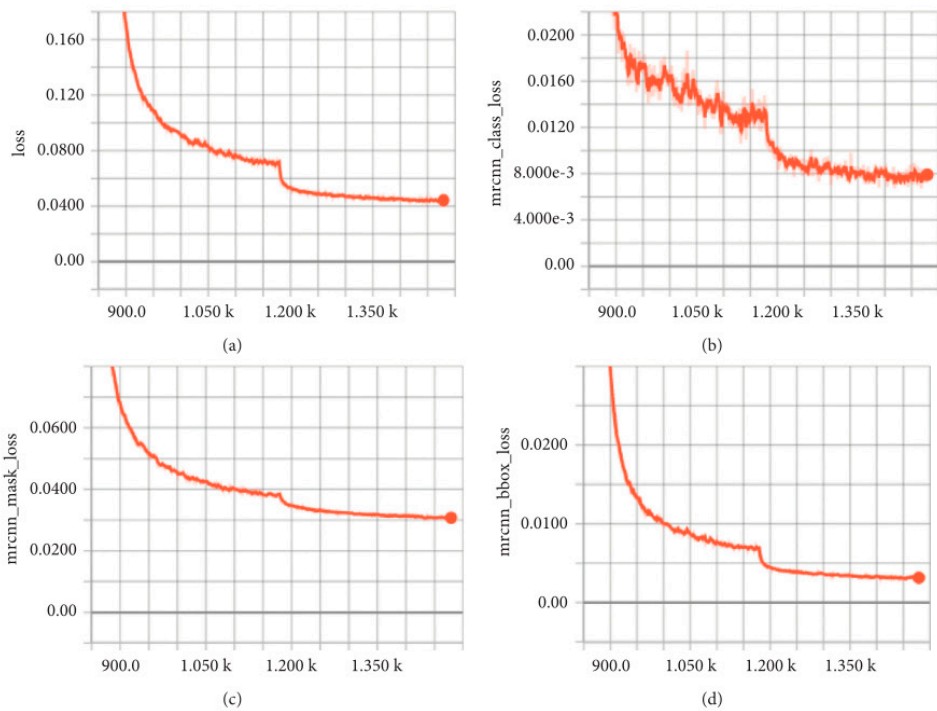

**Figure 12.** Different kinds of loss curves: (**a**) loss; (**b**) class_loss; (**c**) mask_loss; and (**d**) bbox_loss.

**Table 1.** The successful recognition rate.

|       | Recognition | Rate (%) |
| ----- | ----------- | -------- |
| Type1 | 49 | 98 |
| Type2 | 50 | 100 |
| Type3 | 50 | 100 |
| Type4 | 49 | 98 |
| Type5 | 50 | 100 |
| Type6 | 50 | 100 |
| Average | 49.67 | 99.33 |

In Table 1, the recognition rate results are obtained from the six types of switch machines. It should note that each type of switch machine contains 50 images which are randomly selected from the existing dataset built from daily works for the recognition experiment and the following experiments for testing. The reason for selecting 50 images is that those images contain almost all the statuses of switch machines. A recognition is considered successful if and only if all six screws are recognized in the image.

From Table 1, it can be seen that the recognition rate results of the six image groups are 98%, 100%, 100%, 98%, 100%, and 100%, respectively. For Type1, the feature of the screw region is similar to the surrounding region, and some screws cannot be recognized.

For Type4, the whole features of this switch machine are different from the other switch machines. While for the training process, the training number of Type4 is the same as the others, this may lead to the network not being able to recognize this kind of switch machine well. However, the average recognition rate is 99.33%, which indicates that the screw positions of six types of switch machines can be predicted with a high recognition rate. The subjective predicted positions of the screws are shown in Figure 13. From Figure 13, it can be seen that the predicted outlines of the screws are irregular, and the second segment needs to be analyzed.

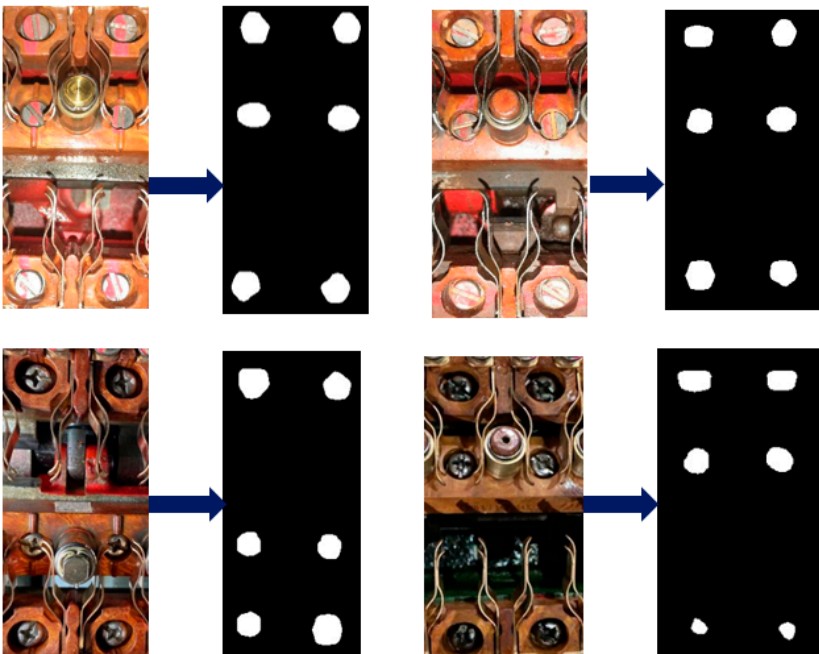

**Figure 13.** Predicted results of the screws from the Mask RCNN.

*3.3. Second Segmentation Results*

The second segmentation process of the screws is shown in Figure 14. The segmentation accuracy evaluation is shown in Table 2.

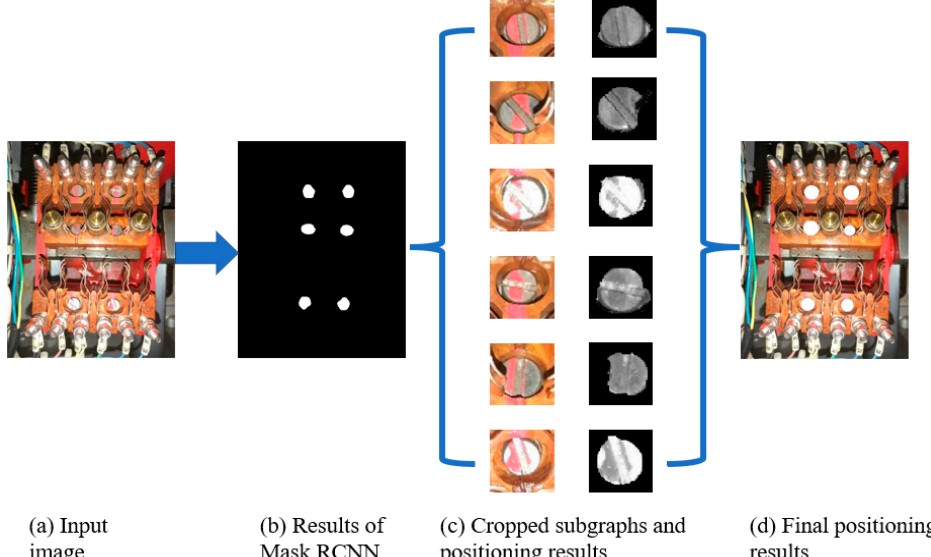

(a) Input image | (b) Results of Mask RCNN | (c) Cropped subgraphs and positioning results | (d) Final positioning results

**Figure 14.** Second segmentation process of the screws.

**Table 2.** Segmentation accuracy evaluation.

|  | Iou | Dice |
|---|---|---|
| Type1 | 0.961 | 0.98 |
| Type2 | 0.951 | 0.975 |
| Type3 | 0.972 | 0.97 |
| Type4 | 0.98 | 0.99 |
| Type5 | 0.982 | 0.991 |
| Type6 | 0.974 | 0.987 |
| Average | 0.97 | 0.982 |

From Table 2, it can be seen that the segmented region obtained via the second segmentation algorithm has deviated little from the actual screw region, which means the segmented region has high reliability.

### 3.4. Experimental Results of the Access Depth

The experimental results of the access depth are shown in Table 3. From Table 3, it can be seen that the average error of the access depth between the proposed algorithm and the actual value is 0.41 mm, which indicates that the proposed algorithm has high accuracy in measuring the access depth between the static contact groups and the dynamic contact groups.

**Table 3.** The experimental results of the access depth.

|  | Proposed Algorithm (mm) | Actual Access Depth (mm) | Error (mm) |
|---|---|---|---|
| Type1 | 6.79 | 7.20 | 0.42 |
| Type2 | 6.84 | 7.30 | 0.46 |
| Type3 | 9.09 | 8.20 | 0.89 |
| Type4 | 6.71 | 6.50 | 0.21 |
| Type5 | 6.83 | 7.00 | 0.17 |
| Type6 | 7.20 | 6.90 | 0.30 |
| Average | 7.24 | 7.18 | 0.41 |

The access depth errors of the six types of switch machines are shown in Figure 15. From Figure 15, it can see that the accurate results are 0.42 mm, 0.46 mm, 0.89 mm, 0.21 mm, 0.17 mm, and 0.3 mm, respectively, which means the proposed algorithm keeps a high level of accuracy in calculating the access depth for the different types of switch machines. It should be noted that the dataset mentioned in Section 3.1 with 2400 images is used for training the networks of Yin et al. [12] and Yolo. As most recent studies have focused on object detection, few of them present the applications for switch machines. Yin et al. [12] attempt to detect the pillar of switch machines with two networks; this is a similar application to our work. Therefore, the algorithm proposed by Yin et al. [12] is selected for the comparison. The experimental results of the proposed algorithm, Yin et al. [12] and Yolo, are shown in Figure 16.

In Figure 16, the access depth recognition rate indicates the successful obtainment of the access depth. From Figure 16, we can see that the proposed algorithm has the highest rate for successfully obtaining the access depth compared to the other algorithms which have 99.3%, 72%, and 75%, respectively. For the access depth accuracy, 98% of the test values of the proposed algorithm are kept within 1 mm. However, the methods proposed by Yin et al. [12] and Yolo have lower accuracy rates for access depth within 1 mm, which are 67% and 68%, respectively. For testing accuracy in the range of 1 mm to 5 mm, the proposed algorithm, Yin et al. [12], and Yolo attain 1.3%, 3%, and 3% accuracy, respectively. For the access depth accuracy of more than 5 mm, our proposed algorithm performs with 0%, and Yin et al. [12] and Yolo perform with 2% and 4% error rates, respectively. It can be seen that the proposed algorithm provides better results than the other two algorithms. In

terms of computational complexity, the proposed algorithm uses 0.3 s per image for testing, and Yin et al. [12] and Yolo use 4 s and 2 s, respectively. Therefore, the computational complexity of the proposed algorithm can be seen as low.

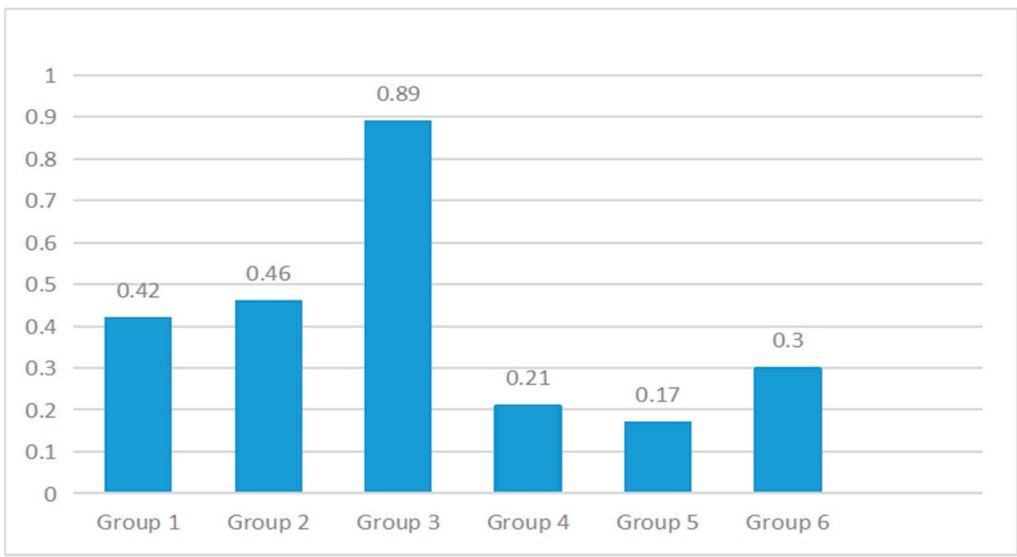

**Figure 15.** Access depth errors of the six types of switch machines.

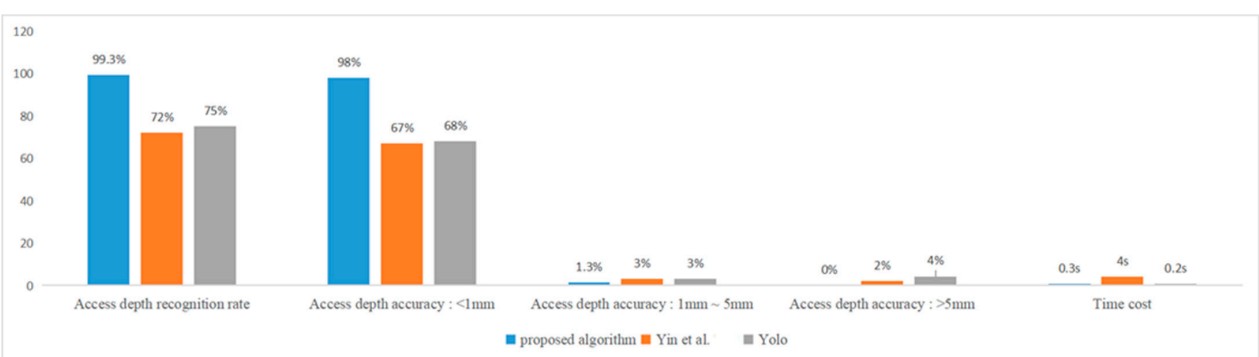

**Figure 16.** Experimental comparisons of the proposed algorithm, Yin et al. [12]'s algorithm, and the Yolo algorithm.

## 4. Conclusions

In this paper, the access depth measurement method for switch machines according to the Mask RCNN is proposed. For accurate testing, firstly, the Mask RCNN is used to obtain the position of the six screws in the dynamic and static contact groups of the switch machine. Then, the second precise segmentation is used to obtain the complete screws. As the deformation of the image will affect the measurement, distortion correction is used to reduce the distortion of the image in the horizontal and vertical directions. Finally, a distance linear fitting and transformation method is used to obtain the accurate access depth for the six types of switch machines. The experimental results show that the quadratic segmentation algorithm can segment small objects accurately. Compared with the direct calibration of the camera, the correction and calculation function can convert the position information into the actual information with a low error margin. The research of algorithms has better promoted the automation of maintenance in the rail transit industry, reducing the work intensity of maintenance personnel, and also providing a new idea for the future of maintenance inspection work. This method can be applied to all types of switch machines, because the key point of selection is the fixing bolt of the automatic switch, which is the same for all switch structures. This method can improve the accuracy, anti-interference,

and universality of the final result for all kinds of complicated situations such as night exposure or tilted pictures.

This method has been trialed by Shanghai Shentong Metro Co., Ltd. for one year; the user report shows the method performs well. For the future, this proposed algorithm needs to be improved mainly in two aspects:

(1) Strict requirements for the framework environment, which are not convenient to change and upgrade.
(2) Before Mask RCNN training, a large number of manual tags need to be generated, and the training efficiency is dependent on the production of these tags.

**Author Contributions:** Conceptualization, L.W., L.K., Z.L., Z.Y. and H.Z.; methodology, L.K., Z.Y. and L.W.; software, L.W., L.K., Z.Y. and H.Z.; validation, L.W., L.K., Z.L., Z.Y. and H.Z.; formal analysis, L.K., Z.Y. and L.W.; investigation, L.W., L.K., Z.L., Z.Y. and H.Z.; data curation, L.W., L.K., Z.L., Z.Y. and H.Z.; writing—original draft preparation, L.W., L.K., Z.L. and Z.Y.; writing—review and editing, L.W. and Z.L. All authors have read and agreed to the published version of the manuscript.

**Funding:** This research received no external funding.

**Conflicts of Interest:** The authors declare no conflict of interest.

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
