# Peer review of "A Low-Complexity Accurate Ranging Algorithm for a Switch Machine Working Component Based on the Mask RCNN"

_applsci, doi:10.3390/app13169424_

Round 1

Reviewer 1 Report

The proposed paper presents an intuitive approach to automatically recognize the status of switch machines using image processing techniques. The paper is mainly well written and its image content definitely helps the understanding of both the problem and the solution.

Remarks:
- Statement needs support or make it a presumption only: the convolutional neural network can extract the characteristics of the object efficiently and give a well prediction.
- References are needed throughout the whole paragraph: With the development of deep learning technology [...]
- There are many doubled whitespaces in the text.
- Wise method as it is, it is unclear how you choose the ideal values for the perspective transformation affine matrix. Please describe it!
- Please describe better the terms of access depth, moving distance. According to Fig. 8. x is the distance of the moving part from the stationary part, but on Fig. 7. x denotes the distance of the two stationary parts (which is denoted with k later). Which one is true? Be more precise on this and correct the figure(s)!
- In the results part, please first describe the evaluation method and the experiments as a short introduction, and only after that, present the details of environment.

Linguistic issues:
- incorrect: correction algorithm, complete the accurate calculation of small target.
- [Since] DUE TO the complexity of the switch machine
- The automatic measurement method is mainly A non-contact detection
- If it [dose] DOES not meet the standard
- incorrect: There is no short improvement based on
- incorrect (even with a point in the middle): Mask was proposed successively On the one hand, R-CNN reduces
- this function [are] IS not [scalability] SCALABLE
- presumably wrong: experimental results are often ?IN?accurate in the ideal environment.
- incorrect: 1. The interesting area detection
- to measure the distance[.] of the access depth.
- [From] COMPARED WITH Fig. 3, we can see that the screws can be segmented precisely.
- incorrect: Due to the image[,] captured [from] WITH A [the] camera or [the] mobile phone, may cause distortion. What causes the distortion here? No subject here.
- incorrect: the image in Fig. 6(b-1) cause the distortion. An image in itself cannot cause distortion. It must be the camera positioning. Or: the image is distorted.
- maybe: the screw regions [] have been obtained with high confidence.
- incorrect: the image will bionship
- incorrect: cropped [as] TO the fixed size of 1280×960.
- The total NUMBER OF images in the date set for training [are] IS 2400
- informal: each kind of the switch machines has 400 images. (Maybe: the training dataset contains 400 images of each kind of switch machine.)
- clause not needed: training[, Loss will experience a temporary rapid decline].
- unintelligible: Polt of the loss functions. Is it plot maybe?
- mistyped: Figure 12. Mask-RCNN calculation results[3.3] Double segmentation positioning results.
- Each group [consists of] contains 50 IMAGES OF one type OF switch machine.
- Incorrect: It should note that the successful target recognition, in Table 1, is defined that all of the six screws in a image are recognized. Maybe: It should be noted that a recognition is considered successful if and only if all six screws are recognized in the image.
- six typeS OF switch machines
- high [accurate] ACCURACY level
- incorrect: has a testing efficiency
- highest rate for SUCCESSFULLY obtaining the access depth [than] COMPARED TO the other algorithmS which [are] HAVE
- incorrect: 98% testing accuracies. Maybe: 98% of the test values
- THE METHODS PROPOSED BY Yin et al [12] and Yolo have [the] lower rateS
- For the testing of accuracy IN THE RANGE OF 1 mm to 5 mm, , the algorithms PROPOSED BY Yin et al [12] and Yolo attain 1.3%, 3% and 3%
ACCURACY respectively.
- [the] OUR proposed algorithm [is] PERFORMS WITH 0%, Yin et al [12] and Yolo [are] PERFORM WITH 2% and 4% ERROR RATES, respectively.
- [For the] IN TERMS OF computational complexity, the proposed algorithm uses 0.3s per image for testing, and Yin et al [12] and Yolo [are] USE 4s and 0.2s respectively.
- Therefore, the computational complexity of the proposed algorithm [is kept at a low level.] CAN BE SEEN AS LOW.
- and the training [effect is related to] EFFICIENCY IS DEPENDENT ON the production of tags.

Author Response

Dear reviewer:

I am very grateful to your comments for this manuscript. Those comments are really valuable and very helpful for improving our paper. We have revised the manuscript mainly according to the comments. The next is our description on revision:

Reviewer #1 Comments Required:

  1. Statement needs support or make it a presumption only: the convolutional neural network can extract the characteristics of the object efficiently and give a well prediction. References are needed throughout the whole paragraph: With the development of deep learning technology [...]

Answer: We have added the reference [5] to indicate the significance of the deep learning, which is marked by highlighting

  1. There are many doubled whitespaces in the text.

Answer: We have moved the doubled whitespaces in the revised manuscript.

  1. Wise method as it is, it is unclear how you choose the ideal values for the perspective transformation affine matrix. Please describe it!

Answer: We have explained how to obtain the transformation affine matrix by the four keys of A B C and D which are shown in Fig. 6. The explanation are marked by highlighting.

  1. Please describe better the terms of access depth, moving distance. According to Fig. 8. x is the distance of the moving part from the stationary part, but on Fig. 7. x denotes the distance of the two stationary parts (which is denoted with k later). Which one is true? Be more precise on this and correct the figure(s)!

Answer: We have rewritten the sentence of how to obtain the access depth, and redraw Fig.7 and Fig.8 by Fig. 8 and Fig. 9 respectively in the revised manuscript. The 6 distance points of the screws are shown in Fig. 8. The access depth is explained in Eq. (4). The moving distance of the screws is shown in Fig. 9. The rewritten part is marked by the highlighting.

  1. In the results part, please first describe the evaluation method and the experiments as a short introduction, and only after that, present the details of environment.

Answer: We have rewritten the experiments. First introduce the experiments and then present the environment. They are marked by the highlighting.

Linguistic Issues:

  1. “correction algorithm, complete the accurate calculation of small target” is rewritten by “Through the secondary segmentation and correction algorithm, the accurate calculation of small target is completed”, which is marked by the highlighting in abstract.

  1. “Since the complexity of the switch machine” is rewritten by “Due to the complexity of the switch machine”, which is marked by the highlighting.

  1. We have removed the sentence of “The automatic measurement method is mainly non-contact detection”.
  2. We have corrected “If it dose not” by “ If it does not”, which is marked by the highlighting.

  1. We have removed the sentence of “There is no short improvement based on RCNN algorithm”. and rewritten “ Fast RCNN,Faster RCNN, Mask R-CNN have been proposed one after another”. They are marked by the highlighting.

  1. We have corrected “this function are not scalability” by “this function is not scalable”.

  1. We have rewritten “the experimental results are often accurate in the ideal environment” as “the proposed algorithm is worked well in the experimental environment”, which is marked by the highlighting.

  1. We have corrected “The interesting area detection” by “The target detection”.

  1. We have rewritten “Due to the image captured with a camera or mobile phone, may cause distortion.” by “Due to the image captured with a camera or a mobile phone may cause the deformation of the switch machine”. They are marked by the highlighting.

10 the image in Fig. 6(b-1) cause the distortion. An image in itself cannot cause distortion It must be the camera positioning. Or. the image is distorted.

Answer: We have rewritten this sentence as “From Fig. 7 (a-1) and Fig. 7 (a-2), it can be seen that the switch machine in Fig. 7 (a) has the deformation in three-dimensional direction. However, in Fig. 7 (b), the deformation is corrected extremely. Similarly, in Fig. 7 (b-1), the deformation of the switch machine is mainly in the horizontal and vertical direction, and the corrected result in Fig. 7(b-2) is almost same as the actual one”. They are marked by the highlighting.

  1. We have rewritten “the screw regions with high confidence have been obtained” as “the screw regions have been obtained with the high confidence”.

  1. We have rewritten “the image will bionship” as “in the image will be calculated”.

  1. We have corrected “cropped as the fixed size” to “ cropped to the fixed size ”.

  1. We have rewritten “The total images in the date set for training are 2400” as “The number of images in the date set for training is 2400”.

  1. We have rewritten “in which each kind of the switch machines has 400 images” as “in which the training dataset contains 400 images of each kind of switch machine”.

  1. unintelligible: Polt of the loss functions. Is it plot maybe?

Answer: It has been corrected as “Different kinds of the loss curves” which is marked by the highlighting.

  1. mistyped: Figure 12.Mask-RCNN calculation results/33] Double segmentation positioning results.

Answer: we have corrected the type of “3.3 Second segmentation results”, which is marked by the highlighting.

17.Each group (consists of] contains 50 IMAGES OF one type OF switch machine.

Answer: It has been rewritten as “In Table 1, the recognition rate results are obtained from the six types of the switch machines. It should note that each type of the switch machine contains 50 images for the recognition experiment and the following experiments for testing. A recognition is considered successful if and only if all six screws are recognized in the image.”. It is marked by the highlighting.

  1. It should note that the successful target recognition, in Table 1, is defined that all of the six screws in a image are recognized. Maybe: lt should be noted that a recognition is considered successful if and only if all six screws are recognized in the image

Answer: It has been rewritten as “In Table 1, the recognition rate results are obtained from the six types of switch machines. It should note that each type of switch machine contains 50 images which is randomly selected from the existing dataset built by daily works for the recognition experiment and the following experiments for testing. The reason of selecting 50 images is that those images almost contain all the status of switch machines. A recognition is considered successful if and only if all six screws are recognized in the image.”. It is marked by the highlighting.

  1. We have rewritten “highest rate for obtaining the access depth than the other algorithms” as “highest rate for successfully obtaining the access depth compared to the other algorithms which have”.

  1. THE METHODS PROPOSED BY Yin et al [12 and Yolo have the] lower rateS. For the testing of accuracy IN THE RANGE OF 1 mm to 5 mm , the algorithms PROPOSED BYYin et al and Yolo attain 1.3%, 3% and 3% ACCURACY respectively.

Answer: It has been rewritten as “However, the methods proposed by Yin et al [12] and YOLO have the lower rates for the access depth accuracy within 1 mm, which are 67% and 68% respectively. For the testing of accuracy in the range of 1 mm to 5 mm, the proposed algorithm, Yin et al [12] and YOLO attain 1.3%, 3% and 3% accuracy respectively”. It is marked by the highlighting.

     21. OUR proposed algorithm (is] PERFORMS WITH 0%, Yin et al [12] and Yolo (are]PERFORM WITH 2% and 4% ERROR RATESrespectively. (For the] IN TERMS OF computational complexity, the proposed algorithm uses 0.3s per image for testing, and Yin et al [12] and Yolo [are] USE 4s and 02s respectively. Therefore, the computational complexity of the proposed algorithm (is kept at a low level.] CAN BE SEEN AS LOW

Answer: It has been rewritten as “Our proposed algorithm performs with 0%, Yin et al [12] and YOLO perform with 2% and 4% error rates respectively. It can be seen that the proposed has a better results than the other two algorithms. In terms of computational complexity, the proposed algorithm uses 0.3s per image for testing, and Yin et al [12] and YOLO use 4s and 02s respectively. Therefore, the computational complexity of the proposed algorithm can be seen as low”. It is marked by the highlighting.

22.“the training effect is related to the production of tags” is written as “ the training efficiency is dependent on the the production of tags”

Thank you again for your time and all the suggestions to our works.

Sincerely yours

Reviewer 2 Report

To change the disadvantages of low efficiency and strong subjectivity of traditional schemes, a low-complexity accurate ranging algorithm is proposed to measure the on-off working parts. Mask-RCNN and interactive iterative method are used in segmentation the ROI accurately. The accurate actual distance is calculated by fitting the linear distance transformation equation, next the secondary segmentation and correction algorithm can complete the accurate calculation of small target. The experimental results explain the performance parameters of novel algorithm that can measure the distance of different working parts, comparing it with two other ones.

Comments:

 1)     Please correct graph presented Fig. 2. Mask R-CNN network structure. There are undefined several arrows, blocks, and text.     

2) In opinion of this reviewer, for better understanding of the presented steps in the segmentation (subsec. 2), the authors should present more images (there are only three ones) for different situations that appear  during the segmentation process.  

 3) Please present the pseudocode algorithm (subsect. 2.5) where you should explain all operations used in your system.

  4)     In opinion of this reviewer, the authors should explain much more theoretical details connected with loss functions used (Fig. 11): loss, class loss, mask loss, bbox loss, and their parameters.

   5)     Please correct the text in fig.17, where should be “Yin et al. [12]”.

 6)   The authors revised several methods that employ deep learning approach (frameworks: 5-12), but they use only paper ref.12 for comparison with proposed method. Please explain, why you used this conference paper (2016) and did not use more recent studies during comparison analysis.

  7)     Please correct text before fig. 17 that should be “The Experimental comparisons of the proposed algorithm, Yin et al. [12] and Yolo are shown in Fig.17.”

   8)     Results presented in fig. 17 are not clear for this reviewer. Did you use the same database for proposed method and for algorithm from ref. 12? Please explain details of experimental setting.

   9)     Please revise and correct reference list, there are two references: 13: 13. Yin J and [13] Zhang C.

 10)  This reviewer did not find into the body of the manuscript the references 12, 13. Please correct this mistake.

Author Response

Dear reviewer:

Reviewer #2

I am very grateful to your comments for this manuscript. Those comments are really valuable and very helpful for improving our paper. We have revised the manuscript mainly according to the comments. The next is our description on revision:

  1. Please correct graph presented Fig. 2. Mask R-CNN network structure. There are undefined several arrows, blocks, and text.     

Answer: We have rewritten the experiments.  First introduce the experiments and then present the environment.  They are marked by the highlighting.

  1. In opinion of this reviewer, for better understanding of the presented steps in the segmentation (subsec. 2), the authors should present more images (there are only three ones) for different situations that appear  during the segmentation process.  

Answer: We have rewritten the experiments.  First introduce the experiments and then present the environment.  They are marked by the highlighting.

  1. Please present the pseudocode algorithm (subsect. 2.5) where you should explain all operations used in your system.

Answer: We have rewritten the experiments.  First introduce the experiments and then present the environment.  They are marked by the highlighting.

  1. In opinion of this reviewer, the authors should explain much more theoretical details connected with loss functions used (Fig. 11): loss, class loss, mask loss, bbox loss, and their parameters.

Answer: We have rewritten the experiments.  First introduce the experiments and then present the environment.  They are marked by the highlighting.

  1. Please correct the text in fig.17, where should be “Yin et al. [12]”.

Answer: We have been modified.

6.The authors revised several methods that employ deep learning approach (frameworks: 5-12), but they use only paper ref.12 for comparison with proposed method. Please explain, why you used this conference paper (2016) and did not use more recent studies during comparison analysis.

Answer: As there is a typical application of the switch machines in ref. 12, we use this paper for comparison.

7.Please correct text before fig. 17 that should be “The Experimental comparisons of the proposed algorithm, Yin et al. [12] and Yolo are shown in Fig.17.”

Answer: we have correct the text, which is marked by the highlighting.

8.Results presented in fig. 17 are not clear for this reviewer. Did you use the same database for proposed method and for algorithm from ref. 12? Please explain details of experimental setting.

Answer: We have explained the database for training which is “It should note that the same dataset mentioned in Section 3.1 with 2400 images is used for training the networks of Yin et al. [12] and YOLO”

9.Please revise and correct reference list, there are two references: 13: 13. Yin J and [13] Zhang C.

Answer: We have revised the reference list, which is marked by the highlighting.

10.This reviewer did not find into the body of the manuscript the references 12, 13. Please correct this mistake.

Answer: We have relisted the references and cited the references into the body.

Thank you again for your time and all the suggestions to our works.

Sincerely yours

Reviewer 3 Report

Review comments

1.       Are these data base, groups of 50 images created by author and team or existing data base? Details need to be given

2.       Why specifically only 50 images / group considered? Basis for this number may be explained.

3.       Any Preprocessing methods for images used? No mention about that.  

4.       Table 1- For Group 1 and 4 – successful target recognition is 98%.  Reasons to be discussed.

5.       Practical application of work done / real time application and future scope may be explained in CONCLUSION.

Author Response

Dear reviewer:

Reviewer #3

I am very grateful to your comments for this manuscript. Those comments are really valuable and very helpful for improving our paper. We have revised the manuscript mainly according to the comments. The next is our description on revision:

  1. The authors revised several methods that employ deep learning approach (frameworks: 5-12), but they use only paper ref.12 for comparison with proposed method. Please explain, why you used this conference paper (2016) and did not use more recent studies during comparison analysis.

Answer: As there is a typical application of the switch machines in ref. 12, we use this paper for comparison.

2.Please correct text before fig. 17 that should be “The Experimental comparisons of the proposed algorithm, Yin et al. [12] and Yolo are shown in Fig.17.”

Answer: we have correct the text, which is marked by the highlighting.

3.Results presented in fig. 17 are not clear for this reviewer. Did you use the same database for proposed method and for algorithm from ref. 12? Please explain details of experimental setting.

Answer: We have explained the database for training which is “It should note that the same dataset mentioned in Section 3.1 with 2400 images is used for training the networks of Yin et al. [12] and YOLO”

4.Please revise and correct reference list, there are two references: 13: 13. Yin J and [13] Zhang C.

Answer: We have revised the reference list, which is marked by the highlighting.

5.This reviewer did not find into the body of the manuscript the references 12, 13. Please correct this mistake.

Answer: We have relisted the references and cited the references into the body.

Thank you again for your time and all the suggestions to our works.

Sincerely yours

Round 2

Reviewer 2 Report

This reviewer did not satisfy with answers of the authors in presented Author's Notes. The authors put the same text “We have rewritten the experiments. First introduce the experiments and then present the environment. They are marked by the highlighting.” as the answers for different comments 1-4 of this reviewer. As result, comments 1 and 3 have not been attended, comments 4 and 6 were attended particularly. Also, this reviewer considers that normal form in authors´ answers is exactly presenting the changes for each comment with detail explications. The authors did not explain well the changes performed in revised manuscript.  

Author Response

Dear reviewer:

Reviewer #2

I'm very sorry that the first revision didn't satisfy you, and thank you very much for your comments on this manuscript. These comments are really valuable and helpful to improve our paper. We revised the manuscript again mainly based on the comments. Next is our second description of the revision:

  1. Please correct graph presented Fig. 2. Mask R-CNN network structure. There are undefined several arrows, blocks, and text.     

Answer: We have corrected the graph in Fig. 2 to make it more clearer.

  1. In opinion of this reviewer, for better understanding of the presented steps in the segmentation (subsec. 2), the authors should present more images (there are only three ones) for different situations that appear  during the segmentation process.

Answer: We have given some explanation and segmentation results in Fig. 14 and Fig. 15 to present the segmentation situations.

  1. Please present the pseudocode algorithm (subsect. 2.5) where you should explain all operations used in your system.

Answer: We have rewritten the text of Fig. 10 with the pseudocode, and given the explanation behind Fig. 10.

  1. In opinion of this reviewer, the authors should explain much more theoretical details connected with loss functions used (Fig. 11): loss, class loss, mask loss, bbox loss, and their parameters.

Answer: We have introduced the four kinds of the losses and explained the loss curves in section 3.2.

  1. Please correct the text in fig.17, where should be “Yin et al. [12]”.

Answer: We have redrawn Fig. 17 and corrected the reference number.

6.The authors revised several methods that employ deep learning approach (frameworks: 5-12), but they use only paper ref.12 for comparison with proposed method. Please explain, why you used this conference paper (2016) and did not use more recent studies during comparison analysis.

Answer: we give some explanations for using ref.12 for comparison, which is “As most studies focus on the object detection recently, few of them give the application of switch machine. Yin et al. [12] attempt to detect the pillar of switch machine with two networks, this is a similar application with our work. Therefore, the algorithm proposed by Yin et al. [12] is selected for the comparison”.

7.Please correct text before fig. 17 that should be “The Experimental comparisons of the proposed algorithm, Yin et al. [12] and Yolo are shown in Fig.17.”

Answer: we have correct the text, which is marked by the highlighting.

8.Results presented in fig. 17 are not clear for this reviewer. Did you use the same database for proposed method and for algorithm from ref. 12? Please explain details of experimental setting.

Answer: We have explained the database for training which is “It should note that the same dataset mentioned in Section 3.1 with 2400 images is used for training the networks of Yin et al. [12] and YOLO”

9.Please revise and correct reference list, there are two references: 13: 13. Yin J and [13] Zhang C.

Answer: We have revised the reference list, which is marked by the highlighting.

10.This reviewer did not find into the body of the manuscript the references 12, 13. Please correct this mistake.

Answer: We have relisted the references and cited the references into the body.

Thank you again for your time and all the suggestions to our works.

Sincerely yours